# Clinical and Radiographic Outcomes of Hip Revision Surgery and Cerclage Wires Fixation for Vancouver B2 and B3 Fractures: A Retrospective Cohort Study

**DOI:** 10.3390/jcm13030892

**Published:** 2024-02-03

**Authors:** Vincenzo Di Matteo, Francesco La Camera, Carla Carfì, Emanuela Morenghi, Guido Grappiolo, Mattia Loppini

**Affiliations:** 1Department of Biomedical Sciences, Humanitas University, 20090 Milan, Italy; drvincenzodimatteo@gmail.com (V.D.M.); carlacarfi@libero.it (C.C.); 2Adult Reconstruction and Joint Replacement Service, Division of Orthopedics and Traumatology, Fondazione Policlinico Universitario Agostino Gemelli IRCCS, Largo Agostino Gemelli 8, 00168 Rome, Italy; 3Fondazione Livio Sciutto Onlus, Campus Savona, Università degli Studi di Genova, 17100 Savona, Italy; francesco.lacamera@humanitas.it (F.L.C.); guido.grappiolo@mac.com (G.G.); 4IRCCS Humanitas Research Hospital, Rozzano, 20089 Milan, Italy; emanuela.morenghi@humanitas.it

**Keywords:** hip, knee, periprosthetic femur fractures, revision arthroplasty

## Abstract

Background: The number of patients presenting with periprosthetic hip fractures has increased in recent decades. Methods: Patients who underwent hip revision arthroplasty procedures for Vancouver type B2 and B3 fractures between 2010 and 2021 were included. The primary intended outcome of this study was to determine the reintervention-free survival rate. The secondary intended outcome was to determine clinical and radiographic assessment outcomes at the time of follow-up, and the correlation between time to surgery and postoperative Harris hip score (HHS). Results: A total of 49 patients with mean age of 71.2 ± 2.3 (37–88) years old were included. Overall, the Kaplan-Meier method estimated a survival rate of 95.8% (CI 84.2% to 98.9%) at one year, 91.1% (CI 77.9% to 96.6%) at two years, and 88.5% (CI 74.4% to 95.1%) at three, and up to 10, years. The mean limb length discrepancy (LLD) improved from −13.3 ± 10.5 (range −39 to +10) mm at the preoperative stage to −1.16 ± 6.7 (range −17 to +15) mm, *p* < 0.001 postoperative. The mean HHS improved from 31.1 ± 7.7 (range 10 to 43) preoperative to 85.5 ± 14.8 (range 60 to 100), *p* < 0.001 postoperative. Postoperative HHS was not affected by preoperative time to surgery. Conclusions: Revision arthroplasty is an effective treatment for Vancouver type B2 and B3 fractures.

## 1. Introduction 

The total hip arthroplasty (THA) is one of the most effective orthopedics procedures, and its use has increased significantly over the past 35 years [1]. The number of THAs performed has increased especially amongst elderly populations, which experience a higher prevalence of osteoporosis and poor bone quality. Consequently, the number of periprosthetic hip fractures (PPHFs) following hip primary arthroplasty procedures is expected to continue to rise in coming years [2,3]. PPHFs can occur from major trauma, low energy trauma, spontaneous fracture, or missed intraoperative fractures that propagate postoperatively. PPHFs are classified as the fourth most common indication for revision surgery in international registries [4], and the third most common in the Swedish Hip Registry. They are slightly less frequent in other locales, such as in the United Kingdom and Australia [5,6]. In France, PPHFs are in the second most common indication for revision surgery with a rate of 11% [7]. The reported incidence varies between 0.07% and 18% for countries globally [4,8,9,10,11]. Several studies have been published in the past investigating the risk factors, postoperative morbidity, mortality risk, and management of PPHFs. The risk factors include age, gender, BMI, component fixation and stem design, surgical approach, avascular necrosis, rheumatoid arthritis, and medical conditions that decrease bone density. The surgical challenges of PPHFs are related to the presence of an implant in the intramedullary canal, reduced bone stock, patient aging, and the presence of osteoporosis or poor bone quality. High levels of surgical planning and technical skills are required to perform revision procedures after PPHFs. These procedures are associated with an increased risk of morbidity, mortality, and higher procedural costs [12]. It is crucial to establish principles of surgical treatment and clinical management in order to provide the greatest chance of satisfactory outcome, reduce the rate of complications, and ensure a high rate of prosthetic survival. As these fractures become more common, they are increasingly treated by trauma surgeons as opposed to arthroplasty surgeons, and this could explain the reason they seem to be increasingly undergoing open reduction and internal fixation (ORIF) procedures instead of revision arthroplasties (RA). However, there is no clear evidence regarding the best approach. PPHFs are grouped into two categories, based on the presence of femoral stem stability or underlying stem loosening. Category 1 includes Vancouver types A, B1 and C. In these fracture types, the implant is fixed to the bone, making these types potentially suitable for ORIF. Category 2 includes Vancouver types B2 and B3 (VB2 and VB3) fractures, in which the implant is not fixed to the bone, and these types are thus potentially suitable for RA [13]. Some studies have already investigated management and outcomes following femoral revision arthroplasty for PPHFs [13,14]. These were national studies with multicenter retrospective cohorts, and some bias affected them. These biases included those surrounding management strategies, types of implant, and number of procedures per surgeon. Therefore, further independent cohort studies are required to confirm these findings. This retrospective study describes a single cohort of patients in a high-volume single center that underwent revision arthroplasty for PPHFs, performed by senior surgeons experienced in joint replacement surgery. The primary intended outcome was to determine the reintervention-free survival rate. The secondary intended outcome was to determine clinical and radiographic assessment outcomes at the time of follow-up, and the potential correlation between time to surgery and functional score (HHS) at the time of follow-up. 

## 2. Materials and Methods

### 2.1. Protocol

The present retrospective observational cohort study included all patients undergoing hip revision arthroplasty following prior PPHFs, performed by senior surgeons experienced in joint replacement surgery, from 1 June 2010 to 31 November 2021 at the IRCCS Humanitas Research Hospital, Italy. All individual participants signed written informed consents for undergoing the surgery and for inclusion in the registry of orthopedic surgical procedures, within the scope of research and improvement of clinical practice. Patients were identified from hospital clinical records using the International Classification of Diseases, Ninth Revision, Clinical Modification (ICD9-CM) procedure codes 00.72 (femoral component revision) and diagnostic code 996.43 (femur PPHFs). Eligibility criteria included all patients aged above 18 years old who sustained an acute PPHF of Vancouver type B2 or B3 requiring emergency hospital admission and femoral stem revision arthroplasty. Participant exclusion criteria included sequelae of periprosthetic infections, non-unions, malignancy, implant fractures, interprosthetic fractures, age of under 18 years old, and less than one year of follow-up. All Vancouver type A, B1, and C fractures treated with ORIF and all interprosthetic femur fractures after total hip and knee arthroplasty were also excluded. Femoral stem revision was performed using uncemented monoblock splined tapered gritblasted stems (Wagner-SL Revision stem; Zimmer Biomet, Warsaw, IN, USA) or uncemented modular stems (Arcos Modular Femoral Revision System; Zimmer Biomet, Warsaw, IN, USA). The demographic characteristics of participants (age at surgery, sex, BMI), etiologies, Vancouver type, time to surgery (days), whether the fracture was around a primary or revision implant, implant fixation (cemented or uncemented), details of operative treatment (revision or revision and fixation), including surgical strategy employed, operating time (minutes), and LOS (length of hospital stay) were obtained from medical file records. Patients were divided into the “primary replacement group” or “revision group”, depending on whether the fracture was around a primary or revision implant, respectively. Patients enrolled were investigated for mechanism of injury: major trauma, minor trauma, or spontaneous fracture. Major trauma included road injuries and falls from height. Minor trauma was defined as a simple fall to the floor. Spontaneous fractures were defined as those that occurred without a fall or any obvious trauma. Medical file records, including clinical and surgical records, follow-up visits, and radiographic evaluations of all patients, were reviewed specifically for this study. All preoperative radiographs of the pelvis and hip from the anteroposterior and lateral views were obtained. All fractures were classified according to the Vancouver system, which incorporates the site of fracture, the stability of the implant, and the surrounding bone stock. All fractures were classified as type A, B, or C, according to fracture location. Type-A fractures involved the greater (AG) or the lesser (AL) trochanter. Type-B fractures occurred in the femoral shaft around the prosthetic stem or just below it and involved the bed supporting the implant. Type-C fractures involved the bone well below the stem tip. Fractures were further subdivided into stable and unstable types according to whether surgical stabilization was necessary. Type B1 fractures were associated with a solidly fixed stem, whereas type B2 and B3 fractures were associated with a loose stem. The quality of host bone is crucial for type B fracture subdivision; in the B1 and B2 types, the surrounding bone stock was adequate. If the femoral component was loose and there was severe bone stock loss, whether caused by generalized osteopenia, osteolysis, or severe comminution, the fracture was classified as type B3 [6,15,16]. The Vancouver system is reported in Table 1. 

Reinterventions at one year, two years, three years, five years, and 10 years were recorded.

Functional recovery was determined through comparing preoperative and postoperative Harris hip scores, while radiographic outcomes were determined by comparing preoperative and postoperative limb length discrepancy–that is, distance between the lesser trochanter and the center of rotation of the hip on a patients’ radiograph. 

Clinical and radiographic assessments were recorded preoperatively in the hospital, and postoperatively during the follow-up control visit. A standardized follow-up protocol was performed during the postoperative control visit, which involved the examiner determining the clinical HHS and evaluating X-rays of the pelvis and hip. 

Incidences of implant complications (dislocation, PPHFs, eterometry, loosening, non-union, periprosthetic infection, and failure of synthesis) within 90 days of surgery were obtained from medical records. Additionally, incidences of medical complications within 90 days of surgery, including myocardial infarction and atrial Fibrillation (A-fib), hospital-acquired pneumonia, venous thromboembolism (VTE), cerebrovascular accident (CVA), and urinary tract infection (UTI), were recorded. Post treatment outcomes, such as mortality rate after discharge from the hospital, were also reported.

#### 2.1.1. Preoperative Planning

For each patient, a standard radiographic work-up was performed routinely, with an antero-posterior view of the pelvis and antero-posterior and lateral views of the hip. A calibration sphere of 30 mm (to reduce magnification errors) was used for all digital templating. Bone stock assessment was mandatory in order to select a suitable implant and plan the revision surgery. The proper center of rotation of the femoral head, the lateral and vertical offset from the greater and lesser trochanter, respectively, and the limb length discrepancy (LLD) were preoperatively identified by the surgeon. Stem and cup sizing was performed on the contralateral side because of the existing implant, PPHFs, and bone fragments. The distal bone that could provide axial and rotational support for the revision prosthesis was identified. The position of the stem and the tip before and after the fracture were identified and compared in order to measure the stem prosthetic lower displacement. This distance was added to the prosthetic center of rotation to calculate the real vertical and lateral offset distances. The distance was marked on the digital X-ray and used to template the revision prosthesis. The most proximal portion of the distal fracture fragment was identified preoperatively by the surgeon, and it was used during surgery as a reference for reamer. The distance between this bony landmark and the center of the femoral head mark on the template (medium neck + 0 mm) was measured. During the surgery, this distance could be replicated to achieve the proper depth of reaming, so that the prosthesis was placed in the correct position upon implantation. The implant sizing and positioning were carefully templated. The revision monoblock stem did not gain fixation by 3-point contact, but instead by its placement in the conical reamed bone bed. There was a tight press fit of the splines along 2–4 cm of diaphyseal bone supporting the implant distal to the end of the fracture [17]. This type of implant achieved stability due to the conical shape of the prosthesis, preventing subsidence, and due to the splines, which prevented axial rotation. 

#### 2.1.2. Surgical Technique

According to surgeon preference, all RA procedures were performed using a standard posterolateral approach with the patient in lateral decubitus. In this analysis, we will focus on Vancouver type B2 and B3 fractures, which are treated similarly in our practice. Many different reconstructive options are available for surgeons. Stem designs include long, cemented stems, fully porous coated stems, tapered, splined, modular or monoblock stems, and proximal femoral replacement prostheses. Currently, in our practice, we use uncemented monoblock splined tapered gritblasted stems (Wagner-SL stem; Zimmer, Warsaw, IN, USA) or uncemented Arcos Modular Femoral Revision Systems (Zimmer Biomet Inc., Warsaw, IN, USA). Surgeon preference and expertise determines the use of monoblock or modular uncemented stems for revision arthroplasty techniques. We followed concepts established by Berry, which involve seeking to rest the implant on distal intact bone and reconstructing the proximal femoral fragments around the implant. Once the femur was exposed, the fracture fragments were used like an osteotomy, and any cement debris and bone fragments were removed from the medullary canal. All soft tissue attachments to the fracture fragments were maintained during the approach so that none of the vascular supply is compromised, which aided lateral blood supply for fracture healing [18].

### 2.2. Statistical Analysis

Data were described as numbers and percentages, if categorical, or via mean, standard deviation, and range, if continuous.

Adherence of the continuous variable to a Gaussian distribution was checked with a Shapiro Wilks test. Changes in continuous variables between pre-surgery and last follow-up were explored with the Wilcoxon test, due to non-Gaussian data distribution.

Association between preoperative time to surgery and HHS was studied with linear regression analysis, and the results were expressed as a Pearson rho correlation coefficient, a regression coefficient with a 95% confidence interval. A rho under 0.3 was considered low, while a value between 0.3 and 0.6 was considered to represent a medium linear association, and a rho over 0.6 indicated a high linear association. A linear regression coefficient described the increase in HHS observed for each day of time from admission to surgery.

Reintervention-free survival time was calculated from surgery date to reintervention date for reintervention patients, or to last contact date for free from reintervention patients. Reintervention-free survival was plotted and represented according to the Kaplan-Meier method. The estimate of reintervention-free survival for any reason was calculated at one year, two years, three years, five years, and 10 years, and the estimates were presented as percentages, with a 95% confidence interval. 

All analyses were performed with Stata version 17. A *p* under 0.05 was considered as significant.

## 3. Results

### 3.1. Selection of Study Population

A total of 90 adult patients (91 hips) were identified as having a procedure code which potentially represented revision surgery for PPHFs. The flow chart that determined patient selection and inclusion is shown in Figure 1. A total of four patients affected by acetabular PPHFs and one patient affected by prosthetic neck fracture were excluded. Among 85 femoral periprosthetic component fractures identified, 17 patients (18 hips) were excluded from management strategy modeling, comprising nine patients (10 hips) with less than 12 months of follow-up and eight patients with incomplete data. Furthermore, 19 additional femur PPHFs were excluded: two femur PPHFs following knee arthroplasty, one femur PPHFs treated with reduction and osteosynthesis (ORIF) instead of revision arthroplasty (RA), three Vancouver type AG, five Vancouver type B1, and seven and one sequelae of non-union and infection, respectively. A total of 49 femur PPHFs were enrolled. Demographic results are reported in Table 2.

### 3.2. Characteristics of Study Population

Most of the revision arthroplasties performed were uncemented (46, 93.9%); only three (6.1%) were cemented revision arthroplasties. The majority of the femoral component revisions (36, 73.5%) were performed with an uncemented monoblock long-stem implant (Wagner-SL Revision stem; Zimmer Biomet, Warsaw, IN, USA), while seven (14.3%) were performed with a modular uncemented stem (Arcos Modular Femoral Revision System; Zimmer Biomet, Warsaw, IN, USA), three (6.1%) using a conical stem (Wagner Cone prosthesis Zimmer Biomet, Warsaw, IN, USA), and three (6.1%) using a cemented stem (MS-30 Zimmer Biomet; Warsaw, IN, USA). Almost all of the surgical cases (48, 98.0%) involved component revision and additional fixation, and one (2.0%) involved only revision arthroplasty. Revision of the reciprocal acetabular component was reported in 13 patients (26.5%). The mean length of stay (LOS) was 7.8 ± 5.4 days (range 3–28). A detailed analysis of operative management is presented in Table 3.

Overall, the Kaplan-Meier method estimated freedom from reintervention at 95.8% (CI 84.2% to 98.9%) at one year and 91.1% (CI 77.9% to 96.6%) at two years. Freedom from reintervention stabilized at 88.5% (CI 74.4% to 95.1%) at three years, five years, and 10 years. There was no difference, according to the data available, in the reoperation rate following the use of uncemented monoblock or uncemented modular stems. Figure 2 shows the Kaplan-Meier curve of reintervention-free survival.

The mean Harris hip score for patients improved from preoperative values of 31.1 ± 7.7 (range 10–43) to postoperative values of 85.5 ± 14.8 (range 60–100) at the time of the last follow-up (*p* < 0.001). The mean limb length discrepancy value (LLD) improved from preoperative values of 14.6 ± 8.7 (range −39 to +10) mm to postoperative values of 5.5 ± 4.0 (range −17 to +15) mm (*p* < 0.001). HHS and LLD variations are reported in Table 4. There was no correlation observed between time to surgery and functional outcomes, such as HHS, obtained at each follow-up control visit (rho −0.31, regression coefficient −0.76 (95%CI −1.59; 0.07), *p* = 0.070).

Postoperative radiographic control and management of AP and axial projections in two examples of VB2 and VB3 fractures is shown in Figure 3 and Figure 4.

The most common postoperative mechanical complication was late dislocation, observed in three (6.1%) patients, including two patients during the first 60 postoperative days and one after eight months. The first two patients were treated with closed reduction and did not require revision arthroplasty (RA). The third patient underwent reduction, fixation, and revision arthroplasty (RA) at eight months for greater trochanter fracture following hip dislocation. One patient reported postoperative eterometry that did not require revision surgery. A total of two patients (4.1%) required debridement and revision surgery in order to remove metal cerclage and k-wire after failure of synthesis. During revision arthroplasty, the rate of intra-operative femur PPHFs was 2.0% (*n* = 1). A female patient, aged over 65 years, sustained an ipsilateral diaphyseal supracondylar fracture during the uncemented femoral stem (Wagner-SL Revision) insertion. She underwent fixation with plate and screws (ORIF) and revision arthroplasty (RA), and did not suffer from any new PPHFs during the follow-up period. 

Following revision arthroplasty, the rate of postoperative femur PPHFs was 0%. 

A total of four (8.2%) patients required readmission to hospital for surgery during the first twelve postoperative months. 

A total of six patients (12.2%) sustained medical complications within 90-day postoperatively: two patients (4.1%) suffered from superficial wound infections, one (2.0%) presented with a urinary tract infection (UTI), one (2.0%) suffered from atrial fibrillation (FA), one (2.0%) from hypoesthesia and sciatic nerve paralysis, and one (2.0%) developed hospital-acquired pneumonia with COPD. The two patients who suffered from superficial wound infections and wound dehiscence were treated successfully with oral antibiotic therapy without debridement and implant retention (DAIR). No non-union was recorded following hip revision arthroplasty (RA) at time of follow-up. Table 5 lists postoperative mechanical and medical complications for the studied cohort. 

At one year postoperatively, the mortality rate was 2.0% (*n* = 1). No patients died within one week postoperatively. One patient died during the first twelve postoperative months. Lastly, at the final follow-up of the 49 patients included in this study, two (4.1%) patients had died: they died at 5 years and 10 years after revision surgery respectively. The mean follow-up time was 63.4 months (12–129 months).

## 4. Discussion

The main findings of the present study were a good reintervention-free survival rate at 10 years, improvements in clinical and radiographic outcomes, and an absence of correlation between time to surgery and functional score at the time of follow-up, following revision arthroplasty (RA) for Vancouver B2 and B3 PPHFs.

PPHFs are a serious complication following primary and revision hip arthroplasties. As the utilization of both primary and revision arthroplasties grows, the number of admissions for this complication is expected to increase. This study investigated management and clinical and radiological outcomes in patients treated for Vancouver type B2 and B3 PPHFs. The goal of surgical treatment of PPHFs is to ensure bone stability and fracture healing and restore limb function. Vancouver B2 and B3 fractures still present challenges, because these fracture types involve the bone next to the femoral stem. The variety of possible intervention methods and implants and their combinations means that no “gold standard” exists. Fixation alone Is too rigid and can result in problems, as some degree of movement is necessary for callus formation, and increasing stiffness of a construct can lead to mechanical fatigue [19]. 

The mortality rate at one year after revision surgery was 2.0% (*n* = 1). Previously reported mortality rates for PPHFs have been subject to variability, ranging from 11.3% to 17.1% at one year. These studies were limited by a small sample size [20,21]. In our center, the awareness of a higher incidence of major systemic complications in this subset of patients alerted the treating surgeon to carry out comprehensive perioperative management, which potentially led to better outcomes [22]. 

The dislocation rate within one year post-surgery for our cohort was 6.1% (*n* = 3). This is not consistent with that of other studies, with incidences of dislocation following revision surgery reported to be approximately 10% [23,24,25]. Different studies reported risk factors for dislocation after revision arthroplasty, such as age, compliance, and neuromuscular conditions [26]. There are also procedure-related factors such as prosthetic component size, geometry, and related position, type of implant (dual mobility implant, large diameter), surgical technique, and muscular and soft tissue tension, which are modifiable [27]. In this study, to prevent dislocation and impingement following revision procedures, prosthetic component orientation for all patients enrolled was determined using the femur first surgical technique according the surgeon’s preferences [28]. 

There is strong evidence that delayed surgery in patients with hip fractures increases mortality [29]; however, the effect of expedited surgery for patients with PPHFs remains unclear. Griffiths et al. reported a significant increase in complications in those with a delay > 72 h [30], while Gibb et al. reported no evidence that the time to surgery was a risk factor for mortality at one year. We found no correlation between time to surgery and functional outcomes such as HHS at each follow-up control visit (rho −0.31, regression coefficient −0.76 (95% CI −1.59; 0.07), *p* = 0.070. The absence of association between time to surgery and mortality grants the surgeon sufficient time to plan the revision surgery [31]. We found that mortality at one year was 2.0% (*n* = 1); this is lower than the rate found in other published studies [21,31], and also lower than the rate for femoral native neck fractures, which have been reported to have a mortality rate of between 14% to 36% by one year after injury [32]. 

The incidence of PPHFs is increasing. They frequently occur in elderly populations and often require revision surgery, which carry a significant systemic insult risk. Abdel et al. noted that intra-operative fractures occur 14 times more often with uncemented stems in females older than 65 years, while postoperative fracture risk is independent of age or gender, but still increases with the use of uncemented stems [4]. This agrees with our result; the only patient who sustained an intraoperative PPHF in our study was a female over 65 years old. Older female patients were at greater risk for fracture probably because of poorer bone quality. The majority of intraoperative fractures were directly related to the implantation process, and occurred during femoral broaching or stem insertion [33]. 

Patients who sustain PPHFs are typically frail and elderly. Although age could not be considered an absolute contraindication for the performance of revision arthroplasty procedures, a multidisciplinary approach between the orthopedic surgeon, anesthesiologist, and hospitalist is recommended. This enables multidisciplinary discussion about the appropriate level of care and the initiation of treatment of preoperative modifiable risk factors. Furthermore, co-management allows risk stratification and the selection of appropriate surgical and anesthetic options [22]. 

This study has several limitations. Firstly, it is a retrospective observational study, and as with any database, the quality of data and missing data may have introduced errors. Secondly, the sample size is relatively small, which limits the generalizability of the findings and prevents significant comparisons between modular (Arcos Modular Femoral Revision System; Zimmer Biomet, Warsaw, IN, USA) and monoclock (Wagner-SL Revision stem; Zimmer Biomet, Warsaw, IN, USA) stems adopted in RA procedures. Thirdly, patients are from a single center, which may have introduced selection bias. The preoperative schedule bias resulted in the selection of patients with stable chronic diseases, no clinical emergencies, and low comorbidities; therefore, the conclusions of the study may be applicable only to patients with low comorbidities. There exists also operative bias related to the surgical techniques used, which were adopted according to the senior surgeon’s experience and preference in joint replacement surgery; these results therefore cannot be generalized to all the surgeons. Fourthly, our hospital is a high-volume single center where elective primary and revision replacement surgeries are performed, and therefore, there was no control group (PPHFs treated only with ORIF); this limitation prevented us from comparing and demonstrating the superiority of RA versus ORIF treatment for patients who sustained a PPHF. Fifthly, the assessment of clinical outcomes was performed through comparing the preoperative and postoperative score of a single questionnaire (HHS). HHS was the clinical index used in our clinical practice; it included patient satisfaction, disability perceived in ADL (Activities of Daily Living), and also the range of motion obtained after surgery. It was the only preoperative questionnaire available for all patients enrolled in this study. Sixthly, the Arcos modular stem is a recent acquisition; it became possible to implant only within the last 5 years, and consequently the number of patients who underwent RA and the follow-up period with this system is lower compared with patients who underwent RA with Wagner SL. Further research is necessary to compare the results of modular and monoclock revision stems.

The main strength of this study is that to the best of our knowledge, so far, this represents the first study that examines only revisions of femoral components affected by type B2 and B3 fractures. Secondly, this is the only study that evaluates reintervention-free survival, improvements in clinical and radiographic outcomes, and the absence of correlation between time to surgery and functional score at the time of follow-up in a high-volume single center. 

## 5. Conclusions

Revision arthroplasty is a safe surgical technique to manage PPHFs of Vancouver types B2 and B3. It is a meticulous and reproducible technique to promote fracture healing and stable implant integration in order to allow patients to return their preinjury functional level. The absence of association between time to surgery and mortality grants the surgeon sufficient time to plan the revision surgery.

## Figures and Tables

**Figure 1 jcm-13-00892-f001:**
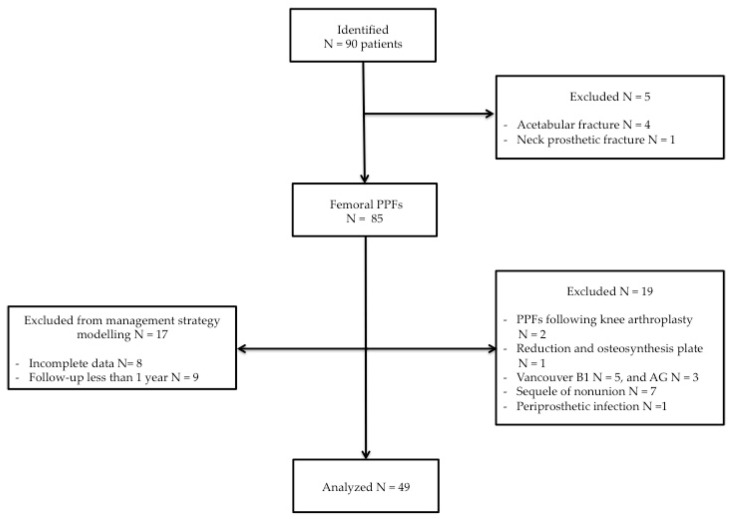
Flow chart of the patient selection and inclusion process.

**Figure 2 jcm-13-00892-f002:**
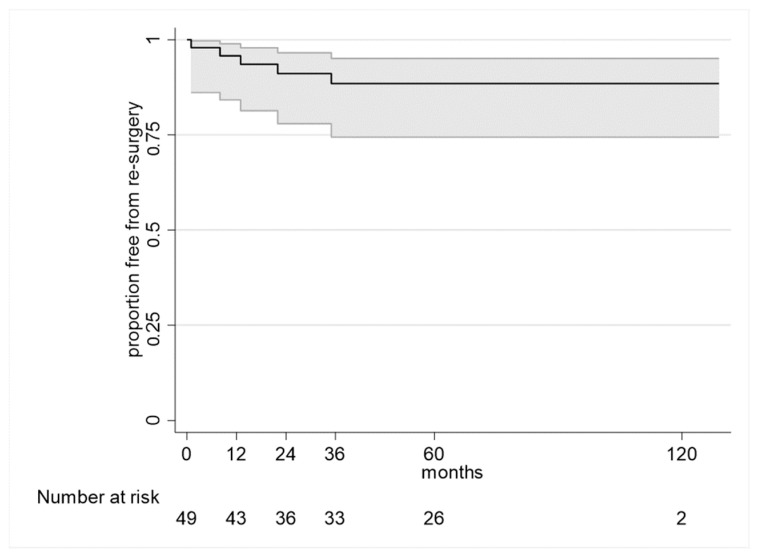
Kaplan-Meier curve of reintervention-free survival with confidence intervals.

**Figure 3 jcm-13-00892-f003:**
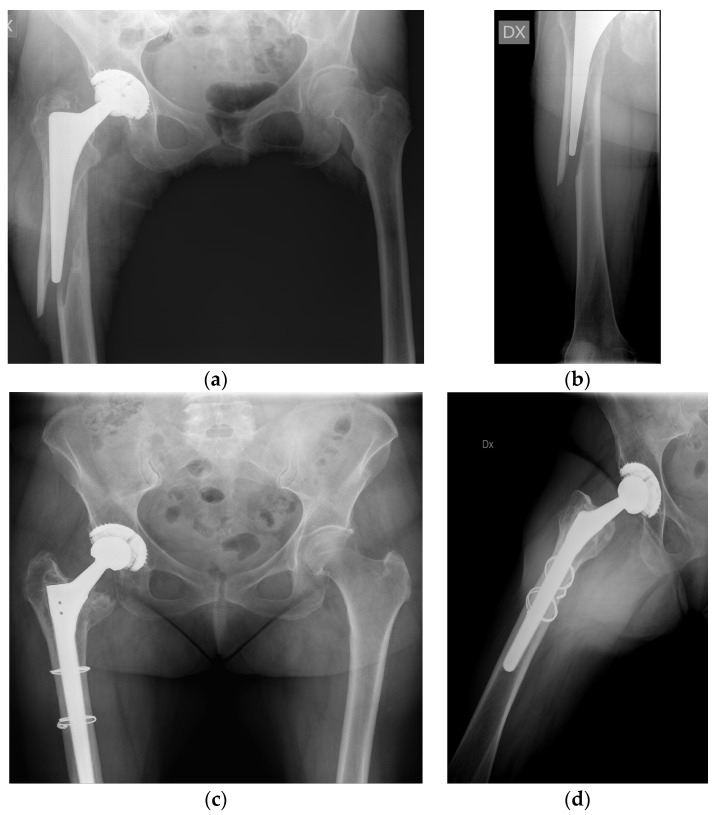
(**a**,**b**) Preoperative radiographs reveal the VB2 fracture of a 70 year old female. (**c**,**d**) Postoperative radiographic results of the same patient managed with stem revision surgery and fixation at 18 months of follow-up. Components used were an adopted monoclock stem (Wagner-SL Revision stem; Zimmer Biomet) and metal cerclages.

**Figure 4 jcm-13-00892-f004:**
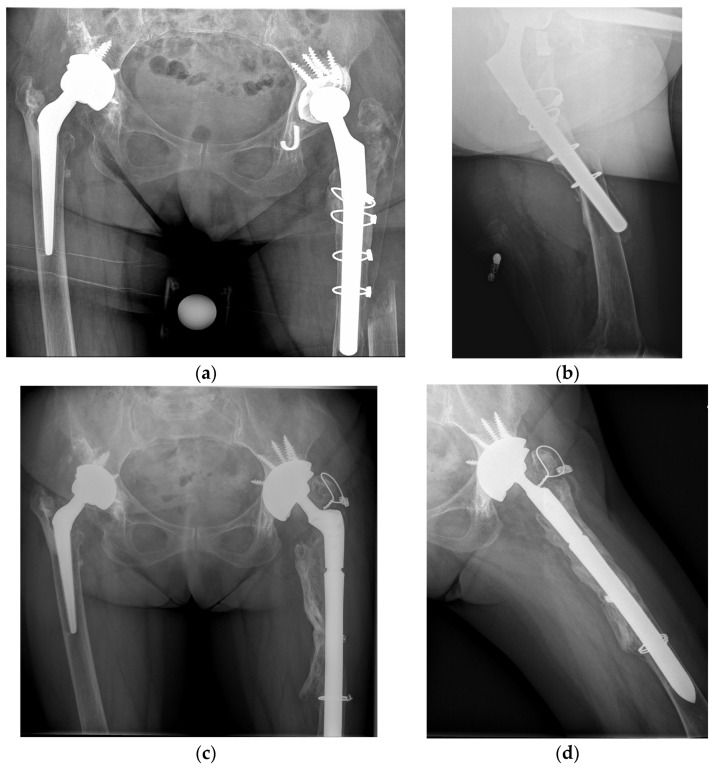
(**a**,**b**) Preoperative radiographs reveal the VB3 fracture of an 83 year old female. (**c**,**d**) Postoperative radiographic results of the same patient managed with total revision surgery and fixation at 18 months of follow-up. Components used were adopted trabecular metal shells (Trabecular Metal Acetabular Revision System; Zimmer Biomet), a modular uncemented stem (Arcos Modular Femoral Revision System; Zimmer Biomet), and iso-elastic polymer and metal cerclages.

**Table 1 jcm-13-00892-t001:** Vancouver classification of postoperative PPHFs.

Classification		Fracture Location
A	AG	Greater trochanter fracture
	AL	Lesser trochanter fracture
B	B1	Fracture around the prosthesis, stem well fixed
	B2	Fracture around the prosthesis, stem is loose
	B3	Fracture around the prosthesis, loose stem, and poor proximal bone stock
C		Fracture distal to tip of stem

**Table 2 jcm-13-00892-t002:** Demographic results.

N	49
Age at surgery	71.2 ± 2.3 (37–88)
Male	15 (30.6%)
Female	34 (69.4%)
Right	29 (59.2%)
Left	20 (40.8%)
PTHA	43 (87.8%)
RTHA	6 (12.2%)
Injury	
Major trauma	6 (12.2%)
Minor trauma	33 (67.3%)
Spontaneous	10 (20.4%)
VB2	44 (89.8%)
VB3	5 (10.2%)

PTHA, primary total hip arthroplasty; RTHA, revision total hip arthroplasty; VB2, Vancouver B2; VB3, Vancouver B3.

**Table 3 jcm-13-00892-t003:** Detail analysis of operative management.

Time to Surgery (Days)	3.9 ± 5.2 (0–30)
Operating time (minutes)	104.6 ± 40.1 (45–224)
RA alone	1 (2.0%)
RA and Fixation	48 (98.0%)
Cerclage alone	44 (91.7%)
Cerclage, screw and K-wire	4 (8.3%)
Stem RA	36 (73.4%)
RTHA	13 (26.5%)
Uncemented stem	46 (93.9%)
Wagner SL revision	36 (78.3%)
Arcos Modular	7 (15.2%)
Wagner Conus	3 (6.5%)
Cemented stem	3 (6.1%)
MS 30	3 (100.0%)
Augments	1 (2.0%)
LOS (days)	7.8 ± 5.4 (3–28)

RA, revision arthroplasty; RTHA, revision total hip arthroplasty; LOS, length of stay.

**Table 4 jcm-13-00892-t004:** Clinical and radiographic outcomes.

	Preoperative	Postoperative	Delta (95% CI)	*p* Value
HHS	31.1 ± 7.7 (range 10 to 43)	85.5 ± 14.8 (range 60 to 100)	54.4 (range 49.6 to 59.2)	<0.001
LLD	14.6 ± 8.7 (range −39 to +10)	5.5 ± 4.0 (range −17 to +15)	12.4 (range 9.9 to 14.9)	<0.001

HHS, Harris hip score; LLD, limb length discrepancy.

**Table 5 jcm-13-00892-t005:** Mechanical and medical complications.

Mechanical Complications	
Dislocation	3 (6.1%)
Intra-operative fracture	1 (2.0%)
Postoperative fracture	0 (0.0%)
Non-union	0 (0.0%)
Periprosthetic infection	0 (0.0%)
Loosening	0 (0.0%)
Eterometry	1 (2.0%)
Failure of synthesis	2 (4.1%)
**Medical complications**	
Superficial Wound infections	2 (4.1%)
UTI	1 (2.0%)
FA	1 (2.0%)
Pneumonia (COPD)	1 (2.0%)
Sciatic nerve palsy	1 (2.0%)
Follow-up (mean (range))	63.4 (range 12–129)

UTI, urinary tract infection; FA, atrial fibrillation; COPD, chronic obstructive pulmonary disease.

## Data Availability

The data supporting reported results can be found in a repository (Zenodo).

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
