# Peer review of "Clinical and Radiographic Outcomes of Hip Revision Surgery and Cerclage Wires Fixation for Vancouver B2 and B3 Fractures: A Retrospective Cohort Study"

_jcm, 2024, doi:10.3390/jcm13030892_

Round 1

Reviewer 1 Report

Comments and Suggestions for Authors

Clinical and radiographic outcomes of hip revision surgery for Vancouver B2 and B3 fractures: a retrospective cohort study

The authors retrospectively assessed the outcomes of 49 patients (mean age 71.2 ± 2.30 (37-88) years) who underwent hip revision arthroplasty for Vancouver B2 and B3, between June 2010 and November 2021. These were when the femoral fracture occured around the primary hip prosthesis, the stem was loose (B2) and there was poor proximal bone stock (B3)

 The overall Kaplan-Meier estimated a survival rate was 95.8% (CI 84.2% to 98.9%) at 1 year,

91.1% (CI 77.9% to 96.6%) at 2 years and

88.5% (CI 74.4% to 95.1%) at 3-10 years.

The mean limb length discrepancy (LLD) improved from – 13.3 ± 10.5 (range - 39 to + 10) mm to a postoperative -1.16 ± 6.7 (range - 17 to + 15) mm, p < 0.001.

The mean Harris Hip Score (HHS) improved from 31.1 ± 7.7 (range 10 to 43) to a post-operative 85.5 ± 14.8 (range 60 to 100), p < 0.001.

There was no evidence that delays to surgery affected the HHS.

The authors concluded that revision arthroplasty was an effective treatment for VB2 and VB3.

Comment:

1)

Overall, I think this is an interesting paper, and reflects a breadth of clinical and surgical experience resulting in a fairly large patient cohort. 

2)

There are a lot of spelling mistakes and missing words in the manuscript. The written English can be improved upon.

3)

Line 33. Make sure that abbreviations actually follow the text. Eg:  Hip Periprosthetic Fractures (PPHFs) does not figure.

Please alter to either  “hip periprosthetic fractures (HPPFs)”, or preferably to “Periprosthetic Hip Fractures (PPHFs)”

4)

Similarly in some places the authors use RA (Revision Arthroplasty), in others RTHA (Revision Total Hip Arthroplasty). Please be consistent, and choose one uniform term throughout the manuscript.

5)

In Table 3 can the authors clarify the difference between the categories?

Surely RA, is the same as RTHA. Or is RTHA also a cup revision?

In which case then surely RA is the same as Stem revision?

Also, how did some patients only have Cerlage alone, or Cerlage and K-wire?  This does not make sense in the context of “non-ORIF” management of B2 and B3 fractures?

I am happy for the authors to include RA plus fixation (cerclage, wires), but not as stand alone categories.

6)

Did any patients also get adjunctive bone grafting?

7)

Do the authors have any specific advice for B3 fractures in particular (compared with B2) from their experience?

Comments on the Quality of English Language

Clinical and radiographic outcomes of hip revision surgery for Vancouver B2 and B3 fractures: a retrospective cohort study

The authors retrospectively assessed the outcomes of 49 patients (mean age 71.2 ± 2.30 (37-88) years) who underwent hip revision arthroplasty for Vancouver B2 and B3, between June 2010 and November 2021. These were when the femoral fracture occured around the primary hip prosthesis, the stem was loose (B2) and there was poor proximal bone stock (B3)

 The overall Kaplan-Meier estimated a survival rate was 95.8% (CI 84.2% to 98.9%) at 1 year,

91.1% (CI 77.9% to 96.6%) at 2 years and

88.5% (CI 74.4% to 95.1%) at 3-10 years.

The mean limb length discrepancy (LLD) improved from – 13.3 ± 10.5 (range - 39 to + 10) mm to a postoperative -1.16 ± 6.7 (range - 17 to + 15) mm, p < 0.001.

The mean Harris Hip Score (HHS) improved from 31.1 ± 7.7 (range 10 to 43) to a post-operative 85.5 ± 14.8 (range 60 to 100), p < 0.001.

There was no evidence that delays to surgery affected the HHS.

The authors concluded that revision arthroplasty was an effective treatment for VB2 and VB3.

Comment:

1)

Overall, I think this is an interesting paper, and reflects a breadth of clinical and surgical experience resulting in a fairly large patient cohort. On this basis, the paper merits publication.

2)

There are a lot of spelling mistakes and missing words in the manuscript. The written English can be improved upon.

3)

Line 33. Make sure that abbreviations actually follow the text. Eg:  Hip Periprosthetic Fractures (PPHFs) does not figure.

Please alter to either  “hip periprosthetic fractures (HPPFs)”, or preferably to “Periprosthetic Hip Fractures (PPHFs)”

4)

Similarly in some places the authors use RA (Revision Arthroplasty), in others RTHA (Revision Total Hip Arthroplasty). Please be consistent, and choose one uniform term throughout the manuscript.

5)

In Table 3 can the authors clarify the difference between the categories?

Surely RA, is the same as RTHA. Or is RTHA also a cup revision?

In which case then surely RA is the same as Stem revision?

Also, how did some patients only have Cerlage alone, or Cerlage and K-wire?  This does not make sense in the context of “non-ORIF” management of B2 and B3 fractures?

I am happy for the authors to include RA plus fixation (cerclage, wires), but not as stand alone categories.

6)

Did any patients also get adjunctive bone grafting?

7)

Do the authors have any specific advice for B3 fractures in particular (compared with B2) from their experience?

Author Response

Manuscript ID: jcm- 2744783

Article Title: Clinical and radiographic outcomes of hip revision surgery for Vancouver B2 and B3 fractures: a retrospective cohort study

Dear Reviewer 1,

Please find attached the revised version of the above manuscript. Your comments have been carefully considered, and implemented as follows. Please note that all changes made in the document have been highlighted in red to facilitate tracking and reading.

  1. We accurately removed redundant sentences, typos, and grammar errors. We also had an English revision of the manuscript.
  2. We correct the manuscript at Line 33, according to your suggestion. 
  3. RA and RTHA have different meanings. RA stands for “Revision Arthroplasty” and it is a general term that indicates a prosthetic revision procedure, which can be a partial exchange (only the femoral stem or the acetabular cup) or the replacement of both the prosthesis components. RHTA, instead, stands for “Revision Total Hip Arthroplasty”, and it means the exchange of both the components, the femoral stem and the acetabular cup. In this study only 13/49 patients underwent RTHA, whereas the other 36/49 underwent stem RA. 
  4. As explained before, RA and RTHA have different meanings. RA is a general term, which includes “stem RA” (stem revision alone) and “RTHA” (stem and cup revisions) as subcategories. How many patients underwent a total revision arthroplasty (RTHA) and a revision of the femoral stem alone is specified in the “stem RA” and “RTHA” line of the table 3. Moreover, there was a pagination error in Table 3, as the rows “cerclage alone” and “cerclage, screw and k-wire” are two subgroups of the row “RA and Fixation “and not as stand-alone categories. Therefore, no fixation alone procedure was performed, but the fixation (such as cerclage/wires) was always accompanied by the prosthetic revision procedure (RA and Fixation).
  5. In this study no patients underwent a bone grafting procedure. We performed the surgery planning to use a revision stem long enough to be thigh press fit along 2-4 cm of diaphyseal bone, distal to the end of the fracture, so the implant was stable and no bone grafting procedures were necessary.
  6. The variety of the possible methods, implants and their combinations means that no “gold standard” exists. From our experience, we can suggest that the use of a modular stem (Arcos Modular Femoral Revision System; Zimmer Biomet, Warsaw, Indiana, US) can be helpful to reduce the risk of LLD.

We thank the Editorial Board and Reviewers for revising our manuscript. We appreciate your and the reviewer’s comments. We hope that the overall quality of the manuscript has improved, and is now amenable for publication in the Journal of Clinical Medicine.

Reviewer 2 Report

Comments and Suggestions for Authors

After reviewing the article "Clinical and radiographic outcomes of hip revision surgery for Vancouver B2 and B3 fractures: a retrospective cohort study," I have identified several important limitations that must be mentioned:

- study is retrospective, which limits its ability to establish causation - however, nothing can be done

- sample size is relatively small (49 patients), which limits the generalizability of the findings (please adjust your discussions accordingly)

- patients are from a single center, which may introduce selection bias - please elaborate on limitations

- there is no control group to compare outcomes against - strong limitation, must be discussed.

- follow-up period varies widely among patients (12-129 months), affecting the study conclusions

- the study type ignores PROMs, overlooking other relevant factors such as patient-reported outcomes or quality of life - please discuss this in the section

- The authors doesn't clearly describe the statistical methods used for data analysis

Author Response

Manuscript ID: jcm- 2744783

Article Title: Clinical and radiographic outcomes of hip revision surgery for Vancouver B2 and B3 fractures: a retrospective cohort study

Dear Reviewer 2,

Please find attached the revised version of the above manuscript. Your comments have been carefully considered, and implemented as follows. Please note that all changes made in the document have been highlighted in red to facilitate tracking and reading.

  • We agree with you, but unfortunately it is an intrinsic limit of this study (retrospective study). In the paragraph “discussion” we took care to explain the several limitations of the study. 
  • The sample size (49 patients) is relatively small, which limits the generalizability of the findings and prevent significantly comparison between modular (Arcos Modular Femoral Revision System; Zimmer Biomet, Warsaw, Indiana, US) and monoblock (Wagner-SL Revision stem; Zimmer Biomet, Warsaw, Indiana, US), the stem adopted in RA procedures. 
  • Third, patients are from a single center, which may introduce selection bias. The preoperative schedule bias could select patients with stable chronic diseases, with no clinical emergencies and low comorbidities, therefore, the conclusion of the study may be applicable only to patients with low comorbidities. The operative bias was related to the surgical technique adopted according to the senior surgeon's experience and preference in the joint replacement surgery; it cannot be generalizable to all the surgeons.
  • Our hospital is a high-volume single center where elective primary and revision replacement surgery are performed, therefore there was  no control group (PPHFs treated with ORIF); this limitation prevented us from comparing and demonstrating the superiority of RA compared to the ORIF treatment for patients who sustained a PPHF.
  • Although the follow-up period varies widely among patients (12-129 months), we established a minimum follow-up period (12 months), which can allow us to better evaluate the clinical and radiographic outcomes. Otherwise, extending the research to patients treated 10 years ago was helpful to increase our sample size.
  • The assessment of the clinical outcome was performed comparing the preoperative and postoperative score of a single questionnaire (HHS). HHS was the clinical index used in our clinical practice, it included the patient satisfaction, the disability perceived in ADL (Activities of Daily Living) and also the range of motion obtained after surgery, it was the only questionnaire preoperative available of all patients enrolled in this study. With regard to the other parameters suggested, unfortunately we didn’t have preoperative data for comparison (which is an intrinsic limit of a retrospective study).
  • The authors have improved and clarified the description of statistical methods used for data analysis as required.

We thank the Editorial Board and Reviewers for revising our manuscript. We appreciate your and the reviewer’s comments. We hope that the overall quality of the manuscript has improved, and is now amenable for publication in the Journal of Clinical Medicine.

Reviewer 3 Report

Comments and Suggestions for Authors

The research paper authored by Vincenzo Di Matteo et al. delves into an intriguing topic. The article presents the results of a retrospective cohort study on clinical and radiographic outcomes of hip revision surgery for 2 Vancouver B2 and B3 fractures. I commend the authors for their dedication and express my appreciation for the chance to evaluate their manuscript. The manuscript is well-crafted, and the data and review protocols are presented clearly and scientifically. Congratulations to the authors on their findings. 

1. There are a couple of redundant sentences, typos, and grammar errors that need to be corrected. I suggest minor english editing.

2. I also suggest that subparagraph 2.1 regarding the protocol be revised and rewritten in a more clearer way. 

The topic is both original and relevant, addressing a critical subject in the field while effectively filling a specific gap.

It provides a unique perspective by presenting findings based on a single center experience, contributing novel insights that differ from other published materials in the subject area.

The conclusions align with the evidence and arguments presented, addressing the main question posed, while the references are deemed appropriate; no additional comments are provided on the tables and figures.

Author Response

Manuscript ID: jcm- 2744783

Article Title: Clinical and radiographic outcomes of hip revision surgery for Vancouver B2 and B3 fractures: a retrospective cohort study

Dear Reviewer 3,

Please find attached the revised version of the above manuscript. Your comments have been carefully considered, and implemented as follows. Please note that all changes made in the document have been highlighted in red to facilitate tracking and reading.

We accurately removed redundant sentences, typos, and grammar errors. We also reviewed and rewrote subparagraph 2.1 regarding the protocol in a clearer way.

We thank the Editorial Board and Reviewers for revising our manuscript. We appreciate your and the reviewer’s comments. We hope that the overall quality of the manuscript has improved, and is now amenable for publication in the Journal of Clinical Medicine.

Round 2

Reviewer 3 Report

Comments and Suggestions for Authors

The performed revision of the manuscript greatly improved the quality of the manuscript. 

Author Response

Dear Review,

I hope this letter finds you well. I am writing to express my gratitude for the effort you invested in reviewing our manuscript entitled Clinical and radiographic outcomes of hip revision surgery and cerclage wires fixation for Vancouver B2 and B3 fractures: a retrospective cohort study” which was submitted to the Journal of Clinical Medicine with Manuscript ID jcm-2744783.

I highly appreciate your constructive feedback and your comments, which have allowed me to enhance the quality of the article, and to address potential limitations. I have carefully considered your recommendations and made the necessary revisions to ensure that the article meets the standards set by JCM.

Thank you for your time and consideration.

Sincerely,

Prof. Mattia Loppini